# Relationship between Autistic Traits and Nutrient Intake among Japanese Children and Adolescents

**DOI:** 10.3390/nu12082258

**Published:** 2020-07-28

**Authors:** Hiromasa Tsujiguchi, Sakae Miyagi, Thao Thi Thu Nguyen, Akinori Hara, Yasuki Ono, Yasuhiro Kambayashi, Yukari Shimizu, Haruki Nakamura, Keita Suzuki, Fumihiko Suzuki, Hiroyuki Nakamura

**Affiliations:** 1Department of Public Health, Kanazawa University Graduate School of Advanced Preventive Medical Sciences, 1-13 Takaramachi, Kanazawa, Ishikawa 920-8640, Japan; ahara@m-kanazawa.jp (A.H.); keitasuzuk@stu.kanazawa-u.ac.jp (K.S.); hiro-n@po.incl.ne.jp (H.N.); 2Kanazawa University Advanced Preventive Medical Sciences Research Center, 1-13 Takaramachi, Kanazawa, Ishikawa 920-8640, Japan; 3Department of Environmental and Preventive Medicine, Graduate School of Medical Science, Kanazawa University, 1-13 Takaramachi, Kanazawa, Ishikawa 920-8640, Japan; toi_fs@yahoo.com (T.T.T.N.); haruki_nakamura@yahoo.co.jp (H.N.); f-suzuki@den.ohu-u.ac.jp (F.S.); 4Innovative Clinical Research Center, Kanazawa University, 13-1 Takaramachi, Kanazawa, Ishikawa 920-8640, Japan; smiyagi@staff.kanazawa-u.ac.jp; 5Department of Neuropsychiatry, Graduate School of Medicine, Hirosaki University, 1 Bunkyocyo, Hirosaki, Aomori 036-8224, Japan; spfy7ff9@wish.ocn.ne.jp; 6Department of Public Health, Faculty of Veterinary Medicine, Okayama University of Science, 1-3 Ikoinooka, Imabari, Ehime 794-8555, Japan; y-kambayashi@vet.ous.ac.jp; 7Department of Nursing, Faculty of Health Sciences, Komatsu University, 1-14 Mukaimotoorimachi, Komatsu, Ishikawa 923-0961, Japan; h_zu@me.com; 8Division of Dental Anesthesiology, Department of Oral Surgery, Ohu University School of Dentistry, 1-31, Misumidou, Tomitamachi, Kohriyama, Fukushima 963-8611, Japan

**Keywords:** ASD, nutrient, children, adolescents

## Abstract

Increased food selectivity among children with autism spectrum disorder (ASD) may lead to nutritional inadequacy. The present study examined differences in nutrient intake between children and adolescents with and without ASD. We utilized cross-sectional data from the ongoing population Shika Town rural Japanese study. The participants were 1276 Japanese pupils and students aged between 7 and 15 years. ASD traits were evaluated using the Autism Spectrum Screening Questionnaire (ASSQ). Nutrient intake was assessed using a food frequency questionnaire. A one-way analysis of covariance (one-way ANCOVA) was performed to compare the mean nutrient intakes between participants with and without ASD traits. A two-way ANCOVA was conducted to compare the mean nutrient intakes among participants with and without ASD traits in different age groups (children and adolescents). The results obtained showed that the intake of carbohydrates and slightly lower intakes of protein, fat, calcium, magnesium, phosphorus, iron, zinc, retinol, vitamin B2, vitamin B12, folic acid, and pantothenic acid were higher among children and adolescents with ASD than among those without ASD. No interactions were observed between the autistic groups and age groups, excluding energy intakes. The present results indicate the importance of screening the nutrient intakes of ASD children and adolescents.

## 1. Introduction

Autism spectrum disorder (ASD) is a neurodevelopmental disorder characterized by “persistent deficits in social communication and social interactions across multiple contexts” and “restricted, repetitive patterns of behavior, interests, or activities” [1]. The estimated prevalence of ASD in children and adolescents worldwide is 1% to 2% [2], and is two to five-fold higher among boys than girls [2,3,4]. The incidence of ASD is increasing [2] and may be attributed to changes in diagnostic criteria, advances in diagnostic techniques, and an awareness or acceptance of ASD [5]. In developed countries, ASD is now recognized as one of the most common serious developmental conditions [6].

Epidemiological studies reported that the prevalence of feeding difficulties among children with ASD ranged between 46–89% [7,8]. The findings of a meta-analysis showed that the prevalence of feeding difficulties was five-fold higher among children with ASD than among typically developing children [8]. One of the most common feeding difficulties is food selectivity (picky eating), characterized by food refusal, eating a limited food repertoire, and the frequent intake of a single food [8,9]. Increased food selectivity may lead to a limited diet and subsequent nutritional inadequacy.

Despite growing evidence for the relationship between ASD and food selectivity in children [8,9], limited information is currently available on the relationship between ASD and nutrient intake. Previous studies compared the nutrient intake of ASD children and typically developing children [8,10]. A recent review reported inconsistencies in the extent and type of nutrient intakes; however, limited macronutrient and micronutrient intakes have been increasingly reported among children with ASD [10]. Additionally, the majority of studies on ASD children used clinical samples with potential concerns about the representativeness of the general population, which may have biased the findings obtained, as well as their interpretations [11]. To exclude this bias, another approach, such as the examination of these relationships within a community-based, representative sample, is needed. Small sample sizes (less than 1000) and age disparities among children were identified as limitations in previous studies. Moreover, few studies investigated whether nutrient intakes among participants with ASD depended on age.

Therefore, the present study was performed to examine differences in nutrient intakes between children and adolescents with and without ASD and whether these differences are dependent on age using a larger and representative community-based sample.

## 2. Materials and Methods 

### 2.1. Participants

We utilized cross-sectional data from the Shika study, an ongoing population-based survey in Shika Town, which is located in a rural area of Ishikawa prefecture, Japan. The town has a population of approximately 20,000 [12]. Data for this analysis were collected between October and November, 2016. Participants were recruited from the Japanese elementary and middle schools in Shika Town. A total of 1335 pupils and students aged between 7 and 15 years were invited, and 1276 ultimately participated (response rate 95.6%). Participation in this study was voluntary. None of the participants received any remuneration for taking part. All guardians of the pupils and students gave informed consent for their inclusion before participation in the study. The present study was conducted in accordance with the Declaration of Helsinki, and the protocol was approved by the Ethics Committee of Kanazawa University (No. 2568).

### 2.2. Screening for ASD

ASD traits were evaluated using the Autism Spectrum Screening Questionnaire (ASSQ). The ASSQ is a parent or teacher-rating scale designed for screening children and adolescents who may have ASD in both clinical and nonclinical settings [13,14]. It is standardized for an age range between 7 and 16 years and allows for screening the broader phenotypes of ASD traits [13,14]. It contains 27 items under 4 domains: social interaction, communication problems, restricted and repetitive behavior, and motor clumsiness. Each item is rated on a 3-point Likert scale: “0” indicating normal, “1” indicating some extent abnormal, and “2” indicating definitely abnormal. Total ASSQ scores range between 0 and 54, with higher scores indicating more severe ASD traits. The validity of the ASSQ has been confirmed with good interrater reliability (r = 0.77) and excellent test-retest reliability (r = 0.96 for parental report and 0.94 for teacher report) [14]. It was translated into the Japanese version by the Ministry of Education, Culture, Sports, Science and Technology with the original authors’ permission [15]. Its reliability and validity for Japanese populations has also been confirmed [16]. An ASSQ cut-off score of 13 was shown to discriminate between clinical and nonclinical samples with 91% sensitivity and 77% specificity [14]. Participants were dichotomized into groups with and without ASD traits by ASSQ scores on the cut-off point.

### 2.3. Nutrient Intake

Nutrient intake was assessed using a food frequency questionnaire. We utilized the Brief self-administered Dietary History Questionnaire (BDHQ). The BDHQ is based on a comprehensive version of the validated questionnaire, the Dietary History Questionnaire (DHQ) [17]. BDHQ-10y and -15y (for Japanese elementary school pupils aged 7–12 years and middle school students aged 13–15 years, respectively) were developed by modifying the adult version. We used BDHQ-10y for elementary school pupils and BDHQ-15y for middle school students. The BDHQ asks about dietary history during the preceding month. Dietary history includes mineral or vitamin supplement use. Answers are based on a Likert scale. Guardians were asked to complete the questionnaire for pupils and students. Pupils and students with the ability to understand their dietary intakes also participated in completing the questionnaire. The intake of energy and 99 nutrients was estimated using a computer algorithm for the BDHQ [18]. Each nutrient intake was reported in terms of energy density (% energy or per 1000 kcal). The validity of the BDHQ has already been confirmed [18,19,20]. Details on the BDHQ are described elsewhere [17,18,20,21].

### 2.4. Characteristics

Guardians completed a general questionnaire on the home environment, lifestyle, and medical history of the pupils and students. The questionnaire included questions on sex; age; height; weight; smoking in the family; exercise time; screen time (television, computer, and mobile phone use); food allergies; and elimination diet. Pupils and students with the ability to understand their condition also participated in completing the questionnaire. Participants were divided into two age groups: children, including elementary school pupils, and adolescents, including middle school students. The evaluation of the weight status was completed using the standardized sex-, age-, and height-specific weight references by the Ministry of Education, Culture, Sports and Technology of Japan [22]. Participants were categorized into two groups: non-overweight/obese group, including underweight (<15th percentile) and healthy (>15th to <75th percentile), and overweight/obese group, including overweight (>75th to <85th percentile) and obese (>85th percentile).

### 2.5. Exclusion Criteria

A previous study reported that more children with ASD than typically developing children took nutritional supplements [23]. The use of nutritional supplements may lead to an excessive intake of micronutrients, such as minerals or vitamins, among ASD participants. Therefore, we excluded participants with high-dose supplement use. Since dietary restrictions, such as gluten-free/casein-free diets by parents or guardians, may influence nutrient intakes, it is difficult to confirm whether differences between ASD participants and typically developing children are a result of dietary restrictions or food selectivity [24]. Therefore, exclusion criteria included those with an elimination diet. Participants with an energy intake of less than 600 kcal or more than 4000 kcal per day, which were over- or under-reporters, were also excluded from the analyses.

### 2.6. Statistical Analysis

Data were summarized using descriptive statistics. Continuous variables were described as means and standard deviations (SD), while categorical variables were described as frequencies and percentages. To compare the mean or percentage of variables between boys and girls, independent sample Student’s *t*-tests for continuous variables and chi-squared tests for categorical variables were used. Differences in characteristics between participants with and without ASD traits were examined using independent sample Student’s *t*-tests for continuous variables and chi-squared tests for categorical variables. Nutrient intake served as the primary dependent variable for the analyses described below. A one-way analysis of covariance (one-way ANCOVA) was performed to compare the mean nutrient intake between the participants with and without ASD traits. ANCOVA was adjusted for covariates, such as sex, age, smoking in the family, exercise time, screen time, weight status, and food allergy. A two-way ANCOVA was performed to compare the mean nutrient intake among participants with and without ASD traits in different age groups (children and adolescents). A simple main effect test was performed for nutrients in which an interaction was observed. Furthermore, to investigate the relationship between individual ASD traits and nutrient intake levels, a multiple linear regression analysis for the intake of each nutrient was performed utilizing forced entry methods with adjustments for covariates. The significance of the differences was defined as *p* < 0.05. Data analyses were performed using the IBM Statistical Package for Social Sciences (SPSS) Version 23.0 (IBM, Armonk, NY, USA). 

## 3. Results

### 3.1. Descriptive Statistics of the Participants

The characteristics, ASD traits, and nutrient intakes of participants are summarized in Table 1. In total, 1108 participants were included in the analyses (48.19% (N = 534) were boys). On average, participants were 10.96 (SD = 2.67) years old. The average ASSQ score was 3.93 (SD = 5.39). ASD traits were detected in 7.40% (N = 82) of participants. Mean exercise time (mean = 1.37), screen time (mean = 4.05), ASSQ score (mean = 4.33), energy intake (mean = 1751.68), and carbohydrate intake (mean = 56.77) were significantly higher in boys than in girls (mean = 1.08, 3.81, 3.61, 1525.18, and 55.50, respectively). The percentage of boys who were overweight/obese was also significantly higher (28.84%) than girls (20.73%). Girls had a higher intake compared to boys for protein (mean = 13.93 vs. mean = 13.46), fat (mean = 29.14 vs. mean = 28.15), natrium (mean = 2636.17 vs. mean = 2505.96), potassium (mean = 1150.82 vs. mean = 1091.36), calcium (mean = 349.71 vs. mean = 338.32), magnesium (mean = 123.92 vs. mean = 119.29), phosphorus (mean = 561.38 vs. mean = 542.77), iron (mean = 3.77 vs. mean = 3.56), β-carotene (mean = 1367.97 vs. mean = 1249.63), vitamin D (mean = 4.53 vs. mean = 4.07), α-tocopherol (mean = 3.55 vs. mean = 3.34), vitamin B1 (mean = 0.39 vs. mean = 0.37), vitamin B2 (mean = 0.66 vs. mean = 0.63), niacin (mean = 6.41 vs. mean = 6.09), vitamin B6 (mean = 0.54 vs. mean = 0.51), vitamin B12 (mean = 3.34 vs. mean = 3.07), folic acid (mean = 140.98 vs. mean = 131.14), pantothenic acid (mean = 3.19 vs. mean = 3.12), vitamin C (mean = 48.25 vs. mean = 43.32), and total dietary fiber (mean = 5.79 vs. mean = 5.47).

### 3.2. Comparison of Nutrient Intakse between Participants with and without ASD Traits

Table 2 summarizes the comparisons of characteristics and nutrient intakes between participants with and without ASD traits. The mean exercise time was significantly shorter in participants with ASD traits (mean = 0.88) than in those without ASD traits (mean = 1.24). In contrast, the mean screen time was significantly longer in participants with ASD traits (mean = 4.45) than in those without ASD traits (mean = 3.89). The percentage of those who were overweight/obese was slightly higher among participants with ASD traits (32.93%) than in those without ASD traits (23.98%). No significant differences were observed in sex, age, smoking in the family, and food allergy between the two groups. The mean intake of carbohydrates was significantly higher among participants with ASD traits (mean = 57.64) than in those without ASD traits (mean = 55.99). In contrast, the mean intake was significantly lower among participants with ASD traits than those without ASD traits for protein (mean = 13.03 vs. mean = 13.76), calcium (mean = 317.79 vs. mean = 346.33), magnesium (mean = 117.67 vs. mean = 122.01), phosphorus (mean = 523.13 vs. 554.75), iron (mean = 3.50 vs. mean = 3.69), zinc (mean = 4.26 vs. mean = 4.40), retinol (mean = 124.54 vs. mean = 147.55), vitamin B2 (mean = 0.60 vs. mean = 0.65), vitamin B12 (mean = 2.88 vs. mean = 3.24), and pantothenic acid (mean = 3.04 vs. mean = 3.17)). Participants with ASD also showed a slightly lower intake of fat (mean = 27.79) and folic acid (128.93) than participants without ASD traits (mean = 28.73 and 136.82, respectively). On the other hand, no other significant differences were observed in the intake of energy, natrium, potassium, copper, manganese, β-carotene, vitamin D, α-tocopherol, vitamin K, vitamin B1, niacin, vitamin B6, vitamin C, and total dietary fiber between the groups.

### 3.3. Comparison of Nutrient Intakes among Participants with and without ASD Traits in Different Age Groups

Table 3 summarizes the comparisons of nutrient intakes among participants with and without ASD traits in different age groups. Participants with ASD traits had a higher carbohydrate intake, but lower intakes of protein, calcium, phosphorus, zinc, retinol, vitamin B2, and pantothenic acid than participants without ASD traits. Participants with ASD traits also had slightly lower intakes of fat, magnesium, iron, and vitamin B12 than participants without ASD traits. A significant interaction was observed among ASD traits and age for energy intakes (children with ASD traits = 1503.23, children without ASD traits = 1408.13, adolescents with ASD traits = 1847.16, and adolescents without ASD traits = 2040.29, respectively). In the simple main effect test on adolescents, the mean energy intake of the participants with ASD traits (mean = 1847.16) was significantly lower than that of the participants without ASD traits (mean = 2040.29). No significant interactions were observed in the intakes of the other nutrients.

### 3.4. Relationships between ASD Traits and Nutrient Intakes

Table 4 summarizes the relationships between ASD traits and nutrient intakes. In a multiple linear regression analysis, ASD traits positively correlated with the carbohydrate intake (C = 1.65). In contrast, ASD traits negatively correlated with the intakes of protein (C = −0.73), calcium (C = −28.54), magnesium (−4.33), phosphorus (C = −31.63), iron (C = −0.18), zinc (C = −0.14), retinol (C = −23.01), vitamin B2 (C = −0.05), vitamin B12 (C = −0.35), and pantothenic acid (C = −0.13) and were negatively associated with the intakes of fat (C = −0.95) and folic acid (C = −7.89).

## 4. Discussion

The primary objective of the present study was to examine the relationship between ASD and nutrient intakes among Japanese children and adolescents. The results obtained showed that Japanese children and adolescents with ASD traits both had higher intakes of carbohydrate but slightly lower intakes of protein, fat, calcium, magnesium, phosphorus, iron, zinc, retinol, vitamin B2, vitamin B12, folic acid, and pantothenic acid than those without ASD traits. Furthermore, we found a positive relationship between individual ASD traits and carbohydrate intakes but a negative relationship between ASD traits and the intakes of protein, fat, calcium, magnesium, phosphorus, iron, zinc, retinol, vitamin B2, vitamin B12, folic acid, and pantothenic acid. These results suggest that ASD traits influence macro- and micronutrient intakes both in childhood and adolescence.

In the most recent meta-analysis of 19 studies [10], Esteban-Figuerola et al. (2019) also found that children with ASD consumed less protein, calcium, phosphorus, vitamin B2, and vitamin B12 than those without ASD. However, children with ASD consumed the same amount of carbohydrates, fat, magnesium, iron, zinc, retinol, folic acid, and pantothenic acid as typically developing children. Hyman et al. (2012) [25] reported that children with ASD aged four to eight in the US consumed a large proportion of carbohydrates; however, they reported children with ASD aged nine to 11 consumed similar amounts with typically developed ones. Siddiqi et al. (2019) [5] showed that dietary carbohydrate intakes in ASD children were slightly higher and fat intakes were slightly lower than the recommended levels in South India; however, comparisons with typically developing children were not performed. Other studies revealed that participants with ASD had significantly lower intakes of magnesium [26], iron [26,27,28], zinc [25,28,29], vitamin A [25,28,29,30], and folic acid [29,31,32] than controls, thereby supporting the present results. Although very few studies have investigated pantothenic acid, decreased concentrations of pantothenic acid in whole blood, serum, or plasma have been found in participants with ASD [33]. It is interesting that, in the aspects of mineral and vitamin intakes, the findings were parallel to those observed in other countries.

The difficulty associated with comparing findings from existing studies on this topic is that studies often used different assessment methods for ASD and nutrient intakes [23]. However, the difference between studies may also be due to age heterogeneity. In the present study, we used a wider range of age groups than previous studies, confirming the results obtained both in children and adolescents with larger samples. Differences between studies may also be attributed to the influence of supplement uses or elimination diets. In the present study, we excluded their influence by excluding participants with these criteria.

Food selectivity is common among ASD children [7,9]. It is generally based on the color, shape, texture, or temperature of food [9]. An inverse relationship has been reported between the diet variety and nutritional adequacy; as the diet variety became more limited, the nutrient intake that fell below the recommended levels increased [32]. Food selectivity may occur for a number of reasons, one of which is sensory sensitivity [34]. A previous study reported that sensory issues were extremely common in children with ASD [35]. The DSM-V includes “hyper or hyporeactivity to sensory input” as a symptom of ASD [1]. Eating is one of the daily life activities that may be negatively affected by sensory sensitivity [36]. Lean protein, fresh fruits, and vegetables are foods characterized by strong flavors and textures, and, thus, children with ASD who exhibit sensory sensitivity may be less likely to accept these foods [37]. Another possible reason for food selectivity is gastrointestinal disturbances [38]. Gastrointestinal disturbances have also been frequently reported among the ASD population, and this potentially affects the dietary intake [38]. In addition to the biological mechanisms described above, core autistic characteristics may play roles in food selectivity. Communication and social engagement have been suggested to play important roles in promoting dietary intakes and increasing dietary diversity [8]. In contrast, the characteristics of deficits in social communications and social interactions in the ASD population may lead to limited dietary intakes. The characteristics of restricted, repetitive patterns of behaviors, interests, or activities may also result in self-restricting habits for food in children with ASD [28,37]. They do not like to change their regular diets and, thus, strongly adhere to their routine diet [39].

An interaction effect was observed between ASD traits and age in terms of energy intakes. This result implies that adolescents with ASD traits consume fewer calories than those without. Adolescents with ASD may consume less than typically developing adolescents because of food selectivity. However, regarding other nutrients, differences in nutrient intakes between the participants with and without ASD traits did not vary among children and adolescents. In terms of a nutrient balance, the excessive or limited nutrient intakes in participants with ASD may continue from childhood to adolescence. Previous findings on food selectivity are consistent with the present results. Kuschner et al. (2015) [40] reported that food selectivity was common in a sample of adolescents with ASD, suggesting that it persists throughout childhood. However, in a large cross-sectional study of children and adolescents with ASD, Beighley et al. (2013) [41] found lower levels of food selectivity in older participants. We did not obtain similar results in the present study, which examined nutrient intakes in children and adolescents with and without ASD traits. Some children with ASD may have chronic excessive or limited nutrient intakes until adolescence.

The higher carbohydrate intakes observed in participants with ASD traits may be related to the higher consumptions of cereals. Among cereals, white rice is largely consumed by the ASD population [5]. In countries where rice is one of the staple foods, such as Japan and South India, a preference for rice could lead to excessive carbohydrate intakes. The lower protein or fat intakes in participants with ASD traits may be attributed to the more limited consumptions of animal products, fish, or pulses [5,42]. Furthermore, the lower micronutrient intakes may be related to the limited consumptions of fruits or vegetables among participants with ASD traits [8]. Among micronutrients, the lower calcium intakes may be associated with reduced intakes of dairy products [10,27,43]. Such dietary patterns related to protein and micronutrient intakes by ASD children and adolescents could be common in many countries.

Although we did not find any significant differences of weight statuses between participants with and without ASD, we found significant differences of nutrient intakes between them. This means nutrient problems in ASD children and adolescents can be easily overlooked, when only the weight status is focused on as a standard nutritional health indicator [8]. Nutrients are key barometers of the health status of children and adolescents. When considering the impacts of chronic nutrient problems, excessive or limited intakes of nutrients among children and adolescents with ASD may place them at higher risks of long-term medical complications. Therefore, assessments of nutrient intakes in ASD children and adolescents need to be included as part of routine screening and for the recommendation of interventions. It would enhance awareness among caregivers regarding this issue. Caregivers should refer to nutritionists to assess the excessive or limited nutrient intakes as well. Nutritional support may also be needed in the form of vitamin or mineral supplementations.

The strengths of the present study include the larger sample size than those in previous studies. To the best of our knowledge, this is the largest cross-sectional study to date that examined the relationship between ASD and nutrient intake in a community-based sample of children and adolescents. This strength allowed direct comparisons of the relationship between children and adolescents. In addition, we conducted a complete enumeration on the children and adolescents who go to all schools in a town. We also collected questionnaires with a high response rate. These strengths allowed us to examine relationships, excluding selection biases—particularly, nonrespondent bias—as much as possible.

However, there are several limitations that need to be addressed. Although the present study had a larger representative community-based sample, the sample was obtained from one rural area in Japan. Additionally, we did not collect information on pupils and students who go to schools for the physically or mentally handicapped outside of the town. Therefore, the results obtained may not be generalizable to all children and adolescents. Other larger-scale studies, including pupils and students who go to these schools as participants, are needed. Furthermore, data on ASD, nutrient intakes, and covariates were based on subjective reports rather than objective measurements. Moreover, these data were obtained from parent or guardian reports, even though they were partly verified by the pupils or students themselves. Hence, they may not have been accurately recorded. In defense of these limitations, we performed a comparatively larger-sample scale study. Another limitation is that the present study did not collect information on the parents’ or guardians’ food preferences. Guardians with a limited food repertoire may not offer a wide variety of food to their family, which may have also limited the nutrient repertoires of the children and adolescents [37]. Detailed information on the home environment needs to be examined in a future study. In addition, we did not compare the nutrient intakes with dietary reference intakes (DRI). Therefore, the risk of absolute nutrient deficiency was not clear. Further studies are needed to compare the nutrient intakes with DRI. We were also unable to establish a causal relationship between ASD traits and nutrient intakes, because the present study was cross-sectional in design. Longitudinal studies are needed to assess the directionality and causality of the relationships.

## 5. Conclusions

Japanese children and adolescents with ASD traits consumed more carbohydrates but slightly less protein, fat, minerals, and vitamins than those with ASD traits across all ages examined. The results of the present study indicate the need to screen the nutrient intakes of ASD children and adolescents in order to reduce the risk of an excessive or limited intake by dietary means. Prospective studies are also required to assess long-term changes in the nutrient intakes of children with ASD into adolescence and adulthood.

## Figures and Tables

**Table 1 nutrients-12-02258-t001:** Descriptive statistics of the participants.

	Total (N = 1108)		Boys (N = 534)		Girls (N = 574)		*p*-Value *
	Number/Mean	%/SD	Number/Mean	%/SD	Number/Mean	%/SD	
Sex (Boys)	534	48.19%	－	－	－	－	－
Age	10.96	2.67	10.97	2.64	10.95	2.69	0.881
Adolescents (13–15 years)	396	35.74%	187	35.02%	209	36.41%	0.629
Smoking in the family	403	36.37%	179	33.52%	224	39.02%	0.057
Exercise (hours/day)	1.22	0.99	1.37	0.99	1.08	0.96	0.000
Screen time (hours/day)	3.93	1.78	4.05	1.88	3.81	1.69	0.024
Overweight/obese	273	24.64%	154	28.84%	119	20.73%	0.002
Food allergy	11	0.99%	5	0.94%	6	1.05%	0.855
ASSQ score	3.96	5.39	4.33	5.51	3.61	5.25	0.026
ASD traits	82	7.40%	43	8.05%	39	6.79%	0.424
Energy (kcal)	1634.34	574.14	1751.68	632.90	1525.18	489.35	0.000
Carbohydrate (%energy)	56.11	5.79	56.77	5.56	55.50	5.93	0.000
Protein (%energy)	13.71	2.05	13.46	1.88	13.93	2.18	0.000
Fat (%energy)	28.66	4.80	28.15	4.72	29.14	4.83	0.001
Natrium (mg/1000 kcal)	2573.42	559.78	2505.96	544.81	2636.17	566.65	0.000
Potassium (mg/1000 kcal)	1122.16	244.88	1091.36	232.25	1150.82	252.92	0.000
Calcium (mg/1000 kcal)	344.22	95.40	338.32	93.80	349.71	96.63	0.047
Magnesium (mg/1000 kcal)	121.69	18.85	119.29	18.05	123.92	19.31	0.000
Phosphorus (mg/1000 kcal)	552.41	93.05	542.77	87.68	561.38	97.00	0.001
Iron (mg/1000 kcal)	3.67	0.69	3.56	0.66	3.77	0.70	0.000
Zinc (mg/1000 kcal)	4.39	0.48	4.37	0.45	4.40	0.52	0.253
Copper (mg/1000 kcal)	0.61	0.08	0.61	0.08	0.61	0.08	0.113
Manganese (mg/1000 kcal)	1.55	0.39	1.54	0.38	1.55	0.40	0.916
Retinol (μg/1000 kcal)	145.85	86.81	144.68	107.21	146.93	62.16	0.666
β-Carotene (μg/1000 kcal)	1310.94	774.46	1249.63	764.02	1367.97	780.39	0.011
Vitamin D (μg/1000 kcal)	4.31	2.50	4.07	2.18	4.53	2.75	0.002
α-Tocopherol (mg/1000 kcal)	3.45	0.74	3.34	0.72	3.55	0.75	0.000
Vitamin K (μg/1000 kcal)	104.52	51.57	102.32	50.83	106.57	52.21	0.171
Vitamin B1 (mg/1000 kcal)	0.38	0.07	0.37	0.06	0.39	0.07	0.000
Vitamin B2 (mg/1000 kcal)	0.65	0.15	0.63	0.15	0.66	0.16	0.002
Niacin (mg/1000 kcal)	6.26	1.56	6.09	1.44	6.41	1.65	0.001
Vitamin B6 (mg/1000 kcal)	0.52	0.12	0.51	0.11	0.54	0.12	0.000
Vitamin B12 (μg/1000 kcal)	3.21	1.53	3.07	1.38	3.34	1.66	0.003
Folic acid (μg/1000 kcal)	136.24	39.43	131.14	39.06	140.98	39.21	0.000
Pantothenic acid (mg/1000 kcal)	3.16	0.56	3.12	0.53	3.19	0.58	0.022
Vitamin C (mg/1000 kcal)	45.87	19.48	43.32	19.01	48.25	19.63	0.000
Total dietary fiber (g/1000 kcal)	5.63	1.22	5.47	1.21	5.79	1.20	0.000

* Chi-squared tests for categorical variables and *t*-tests for continuous variables between the sexes. ASSQ: Autism Spectrum Screening Questionnaire and ASD: Autism spectrum disorder.

**Table 2 nutrients-12-02258-t002:** Comparisons of the characteristics and nutrient intakes between participants with and without ASD traits

	With ASD Traits (N = 82)			Without ASD Traits (N = 1026)			*p*-Value *^2^
	Number/Mean *^1^	%/95%CI		Number/Mean *^1^	%/95%CI		
		Lower	Upper		Lower	Upper	
Sex (boys)	43	52.44%		491	47.86%		0.425
Age	10.91	10.39	11.44	10.96	10.80	11.13	0.880
Adolescents (13–15 years)	26	31.71%		370	36.06%		0.429
Smoking in the family	34	41.46%		369	35.96%		0.320
Exercise (hours/day)	0.88	0.66	1.09	1.24	1.18	1.30	0.001
Screen time (hours/day)	4.45	4.03	4.87	3.89	3.78	3.99	0.006
Overweight/obese	27	32.93%		246	23.98%		0.100
Food allergy	0	0.00%		11	1.07%		0.346
Energy (kcal)	1630.81	1520.52	1741.09	1634.62	1603.75	1665.50	0.948
Carbohydrate (%energy)	57.64	56.38	58.89	55.99	55.64	56.34	0.014
Protein (%energy)	13.03	12.60	13.47	13.76	13.64	13.88	0.002
Fat (%energy)	27.79	26.74	28.83	28.73	28.44	29.03	0.089
Natrium (mg/1000 kcal)	2515.51	2404.14	2626.88	2578.05	2546.87	2609.22	0.290
Potassium (mg/1000 kcal)	1094.26	1042.83	1145.70	1124.39	1109.99	1138.79	0.269
Calcium (mg/1000 kcal)	317.79	297.21	338.36	346.33	340.57	352.09	0.009
Magnesium (mg/1000 kcal)	117.67	113.68	121.67	122.01	120.89	123.13	0.041
Phosphorus (mg/1000 kcal)	523.13	503.45	542.80	554.75	549.24	560.26	0.002
Iron (mg/1000 kcal)	3.50	3.36	3.65	3.69	3.65	3.73	0.018
Zinc (mg/1000 kcal)	4.26	4.16	4.37	4.40	4.37	4.43	0.013
Copper (mg/1000 kcal)	0.61	0.59	0.62	0.61	0.61	0.62	0.574
Manganese (mg/1000 kcal)	1.57	1.48	1.65	1.55	1.52	1.57	0.626
Retinol (μg/1000 kcal)	124.54	105.75	143.33	147.55	142.29	152.81	0.021
β-Carotene (μg/1000 kcal)	1311.52	1147.37	1475.68	1310.89	1264.93	1356.85	0.994
Vitamin D (μg/1000 kcal)	3.91	3.37	4.45	4.35	4.19	4.50	0.128
α-Tocopherol (mg/1000 kcal)	3.37	3.21	3.53	3.45	3.41	3.50	0.348
Vitamin K (μg/1000 kcal)	96.30	85.28	107.33	105.18	102.09	108.26	0.129
Vitamin B1 (mg/1000 kcal)	0.37	0.36	0.39	0.38	0.38	0.39	0.296
Vitamin B2 (mg/1000 kcal)	0.60	0.57	0.63	0.65	0.64	0.66	0.003
Niacin (mg/1000 kcal)	6.04	5.71	6.38	6.27	6.18	6.37	0.193
Vitamin B6 (mg/1000 kcal)	0.51	0.48	0.53	0.53	0.52	0.53	0.221
Vitamin B12 (μg/1000 kcal)	2.88	2.55	3.22	3.24	3.14	3.33	0.046
Folic acid (μg/1000 kcal)	128.93	120.75	137.12	136.82	134.53	139.11	0.069
Pantothenic acid (mg/1000 kcal)	3.04	2.93	3.16	3.17	3.14	3.20	0.039
Vitamin C (mg/1000 kcal)	45.43	41.36	49.51	45.91	44.77	47.05	0.826
Total dietary fiber (g/1000 kcal)	5.53	5.27	5.79	5.64	5.57	5.72	0.410

*^1^ Means for characteristics and estimated marginal means for nutrient intakes. *^2^ Chi-squared tests for categorical variables, *t*-tests for continuous variables, excluding nutrient intake, and an analysis of covariance (ANCOVA) for nutrient intakes adjusted by sex, age, smoking in the family, exercise, screen time, weight status, and food allergies.

**Table 3 nutrients-12-02258-t003:** Comparisons of the nutrient intakes among participants with and without ASD traits in different age groups.

		With ASD Traits (N = 82)			Without ASD Traits (N = 1026)			*p*-Value *^2^		
		Mean *^1^	95%CI		Mean *^1^	95%CI		Between Participants with and without ASD Traits	Interaction among ASD Traits and Age	Between Participants with and without ASD Traits in Each Age Group
			Lower	Upper		Lower	Upper			
Energy (kcal)	Children (N = 712)	1503.23	1374.77	1631.68	1408.13	1360.93	1455.33	0.406	0.014	0.154
	Adolescents (N = 396)	1847.16	1657.31	2037.01	2040.29	1967.61	2112.97			0.046
Carbohydrate (%energy)	Children (N = 712)	58.10	56.55	59.66	56.51	55.94	57.08	0.020	0.929	－
	Adolescents (N = 396)	56.77	54.48	59.06	55.05	54.17	55.93			－
Protein (%energy)	Children (N = 712)	12.71	12.18	13.25	13.43	13.23	13.63	0.003	0.995	－
	Adolescents (N = 396)	13.63	12.84	14.42	14.35	14.05	14.65			－
Fat (%energy)	Children (N = 712)	27.72	26.42	29.01	28.55	28.07	29.03	0.093	0.778	－
	Adolescents (N = 396)	27.90	25.98	29.81	29.06	28.33	29.80			－
Natrium (mg/1000 kcal)	Children (N = 712)	2660.75	2530.33	2791.17	2797.07	2749.14	2845.00	0.602	0.077	－
	Adolescents (N = 396)	2259.61	2066.86	2452.37	2185.72	2111.93	2259.51			－
Potassium (mg/1000 kcal)	Children (N = 712)	1052.62	989.54	1115.70	1082.16	1058.98	1105.34	0.326	0.970	－
	Adolescents (N = 396)	1172.72	1079.49	1265.94	1200.06	1164.37	1235.75			－
Calcium (mg/1000 kcal)	Children (N = 712)	313.07	287.63	338.51	339.65	330.30	349.00	0.012	0.812	－
	Adolescents (N = 396)	326.20	288.61	363.80	358.30	343.91	372.70			－
Magnesium (mg/1000 kcal)	Children (N = 712)	118.21	113.27	123.15	123.42	121.61	125.23	0.085	0.563	－
	Adolescents (N = 396)	116.88	109.58	124.17	119.48	116.68	122.27			－
Phosphorus (mg/1000 kcal)	Children (N = 712)	512.31	488.09	536.52	541.70	532.80	550.59	0.004	0.794	－
	Adolescents (N = 396)	543.00	507.21	578.79	578.14	564.44	591.84			－
Iron (mg/1000 kcal)	Children (N = 712)	3.51	3.33	3.69	3.74	3.67	3.80	0.052	0.436	－
	Adolescents (N = 396)	3.50	3.23	3.77	3.59	3.49	3.70			－
Zinc (mg/1000 kcal)	Children (N = 712)	4.22	4.09	4.34	4.37	4.32	4.42	0.032	0.638	－
	Adolescents (N = 396)	4.35	4.16	4.54	4.45	4.38	4.52			－
Copper (mg/1000 kcal)	Children (N = 712)	0.62	0.60	0.64	0.63	0.63	0.64	0.908	0.168	－
	Adolescents (N = 396)	0.58	0.55	0.61	0.57	0.56	0.58			－
Manganese (mg/1000 kcal)	Children (N = 712)	1.50	1.39	1.60	1.52	1.48	1.56	0.320	0.146	－
	Adolescents (N = 396)	1.71	1.55	1.87	1.59	1.53	1.65			－
Retinol (μg/1000 kcal)	Children (N = 712)	116.89	93.74	140.04	136.11	127.60	144.62	0.021	0.610	－
	Adolescents (N = 396)	138.05	103.83	172.26	168.04	154.94	181.14			－
β-Carotene (μg/1000 kcal)	Children (N = 712)	1131.54	929.94	1333.13	1196.61	1122.53	1270.69	0.639	0.238	－
	Adolescents (N = 396)	1667.51	1369.57	1965.45	1515.73	1401.67	1629.79			－
Vitamin D (μg/1000 kcal)	Children (N = 712)	3.50	2.84	4.17	3.92	3.67	4.16	0.161	0.967	－
	Adolescents (N = 396)	4.67	3.69	5.65	5.11	4.74	5.49			－
α-Tocopherol (mg/1000 kcal)	Children (N = 712)	3.28	3.09	3.48	3.37	3.30	3.45	0.446	0.802	－
	Adolescents (N = 396)	3.55	3.25	3.84	3.59	3.48	3.70			－
Vitamin K (μg/1000 kcal)	Children (N = 712)	88.82	75.23	102.40	98.81	93.82	103.80	0.204	0.739	－
	Adolescents (N = 396)	110.72	90.64	130.80	116.58	108.90	124.27			－
Vitamin B1 (mg/1000 kcal)	Children (N = 712)	0.36	0.35	0.38	0.37	0.36	0.37	0.280	0.682	－
	Adolescents (N = 396)	0.40	0.37	0.42	0.41	0.40	0.42			－
Vitamin B2 (mg/1000 kcal)	Children (N = 712)	0.56	0.52	0.60	0.61	0.60	0.62	0.006	0.885	－
	Adolescents (N = 396)	0.69	0.63	0.74	0.73	0.71	0.75			－
Niacin (mg/1000 kcal)	Children (N = 712)	5.83	5.42	6.25	6.02	5.87	6.17	0.196	0.760	－
	Adolescents (N = 396)	6.43	5.82	7.03	6.73	6.49	6.96			－
Vitamin B6 (mg/1000 kcal)	Children (N = 712)	0.49	0.46	0.52	0.50	0.49	0.51	0.283	0.925	－
	Adolescents (N = 396)	0.55	0.51	0.60	0.56	0.55	0.58			－
Vitamin B12 (μg/1000 kcal)	Children (N = 712)	2.70	2.29	3.11	3.06	2.91	3.21	0.072	0.903	－
	Adolescents (N = 396)	3.23	2.63	3.84	3.55	3.32	3.78			－
Folic acid (μg/1000 kcal)	Children (N = 712)	119.25	109.24	129.26	129.63	125.95	133.31	0.182	0.353	－
	Adolescents (N = 396)	147.82	133.02	162.62	149.71	144.05	155.38			－
Pantothenic acid (mg/1000 kcal)	Children (N = 712)	2.89	2.75	3.03	2.99	2.94	3.04	0.039	0.650	－
	Adolescents (N = 396)	3.33	3.12	3.53	3.48	3.41	3.56			－
Vitamin C (mg/1000 kcal)	Children (N = 712)	42.41	37.41	47.40	42.43	40.59	44.27	0.806	0.813	－
	Adolescents (N = 396)	51.04	43.65	58.42	52.14	49.31	54.97			－
Total dietary fiber (g/1000 kcal)	Children (N = 712)	5.53	5.21	5.85	5.71	5.59	5.83	0.582	0.517	－
	Adolescents (N = 396)	5.54	5.07	6.02	5.53	5.35	5.71			－

*^1^ Estimated marginal means. *^2^ Two-way analysis of covariance (two-way ANCOVA) for the nutrient intakes adjusted by sex, age, smoking in the family, exercise, screen time, weight status, and food allergies.

**Table 4 nutrients-12-02258-t004:** The relationships between the ASD traits and nutrient intakes.

	C	SC	95% CI for C		*p*-Value *
			Lower	Upper	
Energy (kcal)	−3.82	0.00	−118.52	110.88	0.948
Carbohydrate (%energy)	1.65	0.08	0.34	2.95	0.014
Protein (%energy)	−0.73	−0.09	−1.18	−0.28	0.002
Fat (%energy)	−0.95	−0.05	−2.03	0.14	0.089
Natrium (mg/1000 kcal)	−62.53	−0.03	−178.37	53.30	0.290
Potassium (mg/1000 kcal)	−30.13	−0.03	−83.62	23.37	0.269
Calcium (mg/1000 kcal)	−28.54	−0.08	−49.94	−7.15	0.009
Magnesium (mg/1000 kcal)	−4.33	−0.06	−8.49	−0.18	0.041
Phosphorus (mg/1000 kcal)	−31.63	−0.09	−52.09	−11.16	0.002
Iron (mg/1000 kcal)	−0.18	−0.07	−0.34	−0.03	0.018
Zinc (mg/1000 kcal)	−0.14	−0.07	−0.24	−0.03	0.013
Copper (mg/1000 kcal)	−0.01	−0.02	−0.02	0.01	0.574
Manganese (mg/1000 kcal)	0.02	0.02	−0.07	0.11	0.626
Retinol (μg/1000 kcal)	−23.01	−0.07	−42.55	−3.46	0.021
β-Carotene (μg/1000 kcal)	0.63	0.00	−170.10	171.37	0.994
Vitamin D (μg/1000 kcal)	−0.44	−0.05	−1.00	0.13	0.128
α-Tocopherol (mg/1000 kcal)	−0.08	−0.03	−0.25	0.09	0.348
Vitamin K (μg/1000 kcal)	−8.87	−0.05	−20.34	2.60	0.129
Vitamin B1 (mg/1000 kcal)	−0.01	−0.03	−0.02	0.01	0.296
Vitamin B2 (mg/1000 kcal)	−0.05	−0.09	−0.09	−0.02	0.003
Niacin (mg/1000 kcal)	−0.23	−0.04	−0.58	0.12	0.193
Vitamin B6 (mg/1000 kcal)	−0.02	−0.04	−0.04	0.01	0.221
Vitamin B12 (μg/1000 kcal)	−0.35	−0.06	−0.70	−0.01	0.046
Folic acid (μg/1000 kcal)	−7.89	−0.05	−16.40	0.63	0.069
Pantothenic acid (mg/1000 kcal)	−0.13	−0.06	−0.24	−0.01	0.039
Vitamin C (mg/1000 kcal)	−0.48	−0.01	−4.72	3.77	0.826
Total dietary fiber (g/1000 kcal)	−0.11	−0.03	−0.39	0.16	0.410

* Multiple linear regression analysis for the nutrient intakes adjusted by sex, age, smoking in the family, exercise, screen time, weight status, and food allergies.

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
