# Peer review of "Relationship between Autistic Traits and Nutrient Intake among Japanese Children and Adolescents"

_nutrients, 2020, doi:10.3390/nu12082258_

Round 1

Reviewer 1 Report

This study details empirical work on autistic traits in children and adolescents, and whether this affects individual's nutrient intake. 

Overall, the study is well articulated, executed and presented clearly.

It is clear from the results that there are some specific dietary differences between individual’s based on their autistic traits, evidenced by an increased consumption of carbohydrates and a reduced intake of various nutrients (e.g., protein, fat, calcium…) in those with high autistic traits.

The findings give further details of the dietary habits of individuals on (potentially) on the spectrum, and autism cognition and perception could underly these habits.

Minor suggestions

In Section 3.1 line 166 change the format of the in-text mean comparisons. It would be easier to read if each comparison was entered into its own set of brackets; e.g., "Girls had a higher intake compared to boys for protein (mean=13.93 vs mean=13.46)..." That way the reader can compare the mean values directly without having to read a separate section of text.

Major suggestions

In the discussion the section on the implications of reduced nutrient intake is quite brief (line 294-299). Can the authors please elaborate on these implications as it forms a significant part of the research agenda?

I was also expecting to read about how cultural differences may affect the numbers reported between similar studies conducted elsewhere. A cross-cultural comparison would provide useful insights moving forward, to discern what is unique and generalisable from your findings.

Author Response

Response to Reviewer 1 Comments

Thank you very much for your e-mail dated March 18th 2020 with the reviewers’ comments. We are returning herewith the manuscript revised according to your e-mail. We have carefully reviewed the comments and have revised the manuscript accordingly. Our responses are given in a point-by-point manner below. Changes to the manuscript are shown using the Track Changes function.

Hereafter, the comments by the reviewers are shown in bold text.

Response to reviewer #1

  • In Section 3.1 line 166 change the format of the in-text mean comparisons. It would be easier to read if each comparison was entered into its own set of brackets; e.g., "Girls had a higher intake compared to boys for protein (mean=13.93 vs mean=13.46)." That way the reader can compare the mean values directly without having to read a separate section of text.

    In accordance with the reviewer’s comment, we changed the sentences to make it easier to compare (line 167-178). Additionally, we changed the sentences in Section 3.2 in the same way (line 190-197).

  • In the discussion the section on the implications of reduced nutrient intake is quite brief (line 294-299). Can the authors please elaborate on these implications as it forms a significant part of the research agenda?

Accordingly, we added implications in the section (line 307-316).

  • I was also expecting to read about how cultural differences may affect the numbers reported between similar studies conducted elsewhere. A cross-cultural comparison would provide useful insights moving forward, to discern what is unique and generalisable from your findings.

    We agree. We added the explanation about cross-sectional comparison (line 247-251, 256-257, 299-300, 304-306).

I would like to thank the editor, assistant editor and the reviewers for their helpful comments and hope that the revised manuscript is now suitable for publication in Nutrients.

Yours sincerely,

Reviewer 2 Report

Review: Relationship between autistic traits and nutrient intake among children and adolescents

The study is well conducted and findings relevant to a wider audience.

In particular the tables were very informative and well set out.

The key findings of the study are very relevant. The assessments of nutrient intake in ASD children and adolescents need to be included as part of routine screening and for the recommendation of interventions. Nutritional support may also be needed in the form of vitamin or mineral supplementation.

While it is small issue, it is interesting that B2(Riboflavin) was low for the students with ASD but not B1 (Thiamine) given that both B1 and B2 occur in some of the same foods, such as eggs. In Japan is there fortification of B1 into the rice to increase the level of B1 in the diet?

The researcher acknowledged that the Japanese setting may influence the outcomes of the study. Japan unlike other parts of the world is not deficient in iodine, with reported links between low gestational iodine levels and higher rates of ASD in countries with iodine deficiency (see review by Hay, I.; Hynes, K.L.; Burgess, J.R. Mild-to-Moderate Gestational Iodine Deficiency Processing Disorder. Nutrients 2019, 11, 1974.).  

There are some minor changes required. In the title, where the study was conducted needs to be included. For example: Relationship between autistic traits and nutrient intake among Japanese children and adolescents. In the abstract information on the Shika study, needs to be included.  For example: We utilized cross-sectional data from the ongoing population Shika town rural Japanese study. The participants were etc   .

I congratulate the researchers for a well conducted and informative study.  

Author Response

Response to Reviewer 2 Comments

Thank you very much for your e-mail dated March 18th 2020 with the reviewers’ comments. We are returning herewith the manuscript revised according to your e-mail. We have carefully reviewed the comments and have revised the manuscript accordingly. Our responses are given in a point-by-point manner below. Changes to the manuscript are shown using the Track Changes function.

Hereafter, the comments by the reviewers are shown in bold text.

Response to reviewer #2

  • While it is small issue, it is interesting that B2(Riboflavin) was low for the students with ASD but not B1 (Thiamine) given that both B1 and B2 occur in some of the same foods, such as eggs. In Japan is there fortification of B1 into the rice to increase the level of B1 in the diet?

We also think it is interesting that B2 was low for the students with ASD but not B1.

It is not common to eat B1 fortified rice as it is expensive and not available at a local shop in Japan. Some people eat brown rice which has more B1.

We will think about investigating the reason for future research.       

  • The researcher acknowledged that the Japanese setting may influence the outcomes of the study. Japan unlike other parts of the world is not deficient in iodine, with reported links between low gestational iodine levels and higher rates of ASD in countries with iodine deficiency (see review by Hay, I.; Hynes, K.L.; Burgess, J.R. Mild-to-Moderate Gestational Iodine Deficiency Processing Disorder. Nutrients 2019, 11, 1974.).

We appreciate informative comments. According to the instruction, we added some explanations about differences among countries (line 299-300). We are planning to conduct research about the iodine intake and urinary iodine levels among children this year. The paper introduced by the reviewer will be useful for our future research.

  • There are some minor changes required. In the title, where the study was conducted needs to be included. For example: Relationship between autistic traits and nutrient intake among Japanese children and adolescents. In the abstract information on the Shika study, needs to be included. For example: We utilized cross-sectional data from the ongoing population Shika town rural Japanese study. The participants were etc.

In accordance with the reviewer’s comment, we added the information about the place where the study was conducted (line 3, 31-33, 81, 235).

I would like to thank the editor, assistant editor and the reviewers for their helpful comments and hope that the revised manuscript is now suitable for publication in Nutrients.

Yours sincerely,